# DeepBiomarker: Identifying Important Lab Tests from Electronic Medical Records for the Prediction of Suicide-Related Events among PTSD Patients

**DOI:** 10.3390/jpm12040524

**Published:** 2022-03-24

**Authors:** Oshin Miranda, Peihao Fan, Xiguang Qi, Zeshui Yu, Jian Ying, Haohan Wang, David A. Brent, Jonathan C. Silverstein, Yu Chen, Lirong Wang

**Affiliations:** 1Department of Pharmaceutical Sciences, Computational Chemical Genomics Screening Center, School of Pharmacy, University of Pittsburgh, Pittsburgh, PA 15206, USA; osm7@pitt.edu (O.M.); pef14@pitt.edu (P.F.); xiq24@pitt.edu (X.Q.); 2Department of Pharmacy and Therapeutics, School of Pharmacy, University of Pittsburgh, Pittsburgh, PA 15206, USA; zey1@pitt.edu; 3Department of Internal Medicine, University of Utah, Salt Lake City, UT 84132, USA; jian.ying@hsc.utah.edu; 4Language Technologies Institute, School of Computer Science, Carnegie Mellon University, Pittsburgh, PA 15213, USA; haohanw@cs.cmu.edu; 5Department of Psychiatry, Western Psychiatric Institute and Clinic, University of Pittsburgh Medical Center, Pittsburgh, PA 15213, USA; brentda@upmc.edu; 6Department of Biomedical Informatics, School of Medicine, University of Pittsburgh, Pittsburgh, PA 15213, USA; 7Eli Lilly and Company, Lilly Corporate Center, Indianapolis, IN 46225, USA

**Keywords:** PTSD, real-world evidence, deep learning, biomarker identification

## Abstract

Identifying patients with high risk of suicide is critical for suicide prevention. We examined lab tests together with medication use and diagnosis from electronic medical records (EMR) data for prediction of suicide-related events (SREs; suicidal ideations, attempts and deaths) in post-traumatic stress disorder (PTSD) patients, a population with a high risk of suicide. We developed DeepBiomarker, a deep-learning model through augmenting the data, including lab tests, and integrating contribution analysis for key factor identification. We applied DeepBiomarker to analyze EMR data of 38,807 PTSD patients from the University of Pittsburgh Medical Center. Our model predicted whether a patient would have an SRE within the following 3 months with an area under curve score of 0.930. Through contribution analysis, we identified important lab tests for suicide prediction. These identified factors imply that the regulation of the immune system, respiratory system, cardiovascular system, and gut microbiome were involved in shaping the pathophysiological pathways promoting depression and suicidal risks in PTSD patients. Our results showed that abnormal lab tests combined with medication use and diagnosis could facilitate predicting SRE risk. Moreover, this may imply beneficial effects for suicide prevention by treating comorbidities associated with these biomarkers.

## 1. Introduction

Each year, about 80 million people worldwide commit suicide, and suicide is the tenth leading cause of death in the United States [1]. Because the suicide rate has continued to rise since 1999, the World Health Organization (WHO) has prioritized the reversal of this increase in the suicide rate [2]. Recently, in the US, suicide was identified as one of the reasons, along with drug overdose, alcohol-abuse and organ system diseases, for increasing mortality among the midlife population [3]. It has been proven that suicide prevention strategies have made a significant contribution in reducing suicide rates. Some have emphasized the importance of understanding depression and suicidal tendencies. The early identification of individuals at risk for developing the neuro-psychiatric disorders coupled with SREs is a major clinical and public health challenge. However, a suicide prevention plan consumes numerous resources, such as labor, time, and money, which limits the application to all patients. Therefore, a resource-saving prediction procedure for patients is urgently needed.

The risk of suicide-related events (SREs) is significantly increased in post-traumatic stress disorder (PTSD) patients [4]. A longitudinal study suggested that in women and men diagnosed with PTSD, suicide rates were 6.74 and 3.96 times higher than in those without PTSD, respectively, and 53.7% of suicides could be attributed to PTSD in this population [5]. PTSD is a disorder that can trigger symptoms that include amnesia, alterations in personal thoughts and feelings, attempts to avoid trauma-related cues, elevated fight-or-flight response, mental or physical distress to trauma-related cues, intrusive, recurrent recollections, dissociative episodes of reliving the trauma (“flashbacks”), and nightmares [6]. People may develop PTSD after suffering a traumatic event such as child abuse, traffic collisions, warfare, sexual assault, or other kinds of threats. These patients usually begin to show symptoms within the first three months after the trauma. However, the symptoms will most often not be recognized and diagnosed until several years after the traumatic event. Although most people will present some level of stress response quickly after encountering the traumatic event, these symptoms must persist long enough (for more than one month) and lead to vital dysfunction or clinical levels of distress to be classified as PTSD. If the symptoms continue for less than a month, it is considered an Acute Stress Disorder (ASD) [6]. However, standard treatment options, such as pharmacological interventions or psychotherapy, have only been partly successful and continue to show considerable variation of effectiveness among these high-risk patients. Alternative techniques, such as non-invasive brain stimulation (NIBS) including repetitive transcranial magnetic stimulation (rTMS) and transcranial direct current stimulation (tDCS), and their potential to treat other pathways of PTSD and other neuropsychiatric disorders have been explored [7,8,9].

Various studies suggest that PTSD is related to dysfunctions in numerous biological systems. For example, it has been associated with alterations in the ventromedial prefrontal cortex (vmPFC) brain structure and its subregions, abnormalities in fear conditioning neural circuits, and smaller lesion volume. Functional alterations of the frontal lobe circuitry affect fear conditioning or emotional regulatory mechanisms, which may contribute to the overall etiology of anxiety-related disorders like PTSD, panic disorders, or phobias [8,10]. According to corroborated literature, PTSD has also been strongly linked to inflammation. Increased pro-inflammatory cytokines such as IL-1β, IL-6, IFN-γ, and TNF-α and anti-inflammatory cytokines such as TGF-β were found in the serum of patients with PTSD and may be correlated with the severity of PTSD. Most patients with PTSD have insufficient integration of a trauma memory, due to unwanted changes in the hippocampal-cortical memory networks, thus forming disorganized and incomplete intrusive memories. In addition, stress hormones like cortisol activate the tryptophan (TRP)-kynurenine (KYN) metabolic pathway, which synthesizes ligands such as kynurenines that transform into a palette of small bioactive molecules with oxidant, antioxidant, neurotoxic, neuroprotective, and/or immunomodulatory properties, affecting sleep and formation of intrusive memories in these high-risk patients [11,12,13,14,15]. Epigenetic modifications affecting regulation of amine, glucocorticoid, and serotonin metabolism production are pivotal in the pathogenesis and development of psychiatric disorders characterized by anxiety and suicide [16]. Currently identified biomarkers can be accumulated from physiological and neuro-endocrine responses [17], gene-expression profiles [18], and peri-traumatic responses [19,20]. This suggests that PTSD is associated with an array of multimodal risk indicators, which may be exacerbated by SREs [21]. Despite these findings, the research to date has failed to reveal translational and personalized biomarkers using EMR data (both clinical and non-clinical) for SRE risk prediction. To determine whether a proposed etiological mechanism is causal for the development of PTSD with SREs, applications of analytic technologies to data-mine the EMR data from these patients can be a promising avenue.

Deep learning/data mining algorithms can translate data into information for hypothesis generation through deep hierarchical feature construction to capture long-range dependencies in EMR data [22]. Recently, a variety of deep learning techniques and frameworks have been applied to information extraction, representation learning, outcome prediction, phenotyping, and de-identification [23,24,25,26,27] and have yielded better performance than traditional methods and required less time-consuming preprocessing and feature engineering. Specifically, deep learning techniques learn optimal features directly from the data itself, without any human guidance, allowing for the automatic discovery of latent data relationships that might otherwise be unknown or hidden [28]. Current biological research has identified group-level risk factors for PTSD and/or SREs, which may not accommodate multi-modal information (lab tests, medication use, diagnosis) [29]. Since deep learning models overcome this shortcoming, they may help in establishing more putative biomarkers.

In order to fill these gaps in biomarker research for PTSD, we leverage the merits of deep learning techniques for SRE prediction. The goal of our present work is to evaluate the EMR data from PTSD patients and to identify potential biomarkers to predict SREs. To accomplish this goal, we built a deep-learning based model, DeepBiomarker, through modification of an established deep-learning framework, Pytorch_EHR [30]. In the DeepBiomarker, we use diagnosis, medication use, and lab tests as the input, implement data augmentation technologies to improve the model performance, and integrate a perturbation-based approach [31] for risk factor identification.

## 2. Method

### 2.1. Data Source

We examined the data from January 2004 to October 2019 from the Neptune system at the University of Pittsburgh Medical Center (UPMC), which manages the use of patient electronic medical records from the UPMC health system for research purposes (rio.pitt.edu/services) [32]. The database includes demographic information, diagnoses, encounters, medication prescriptions, prescription fill history, and laboratory tests. We used the SREs after the diagnosis of PTSD. The PTSD patients were identified by ICD9/10 codes (‘309.81’, ‘F43.10’, ‘F43.11’, and ‘F43.12’), and the SREs were also identified by ICD9/10 codes (See Appendix B) [33].

### 2.2. Data Preparation

For each PTSD patient, we aimed to predict whether the patient would have SREs within next 3 months, given the history of EMRs. In this study, we defined the cases and controls as follows: At any encounter, a PTSD patient who had a record of SREs within the following 3 months was defined as a case, while no records of SREs within the following 3 months was defined as a control. The date of this encounter was the index date. For a patient with multiple encounters satisfying the criteria of control, only the last encounter was included to mimic the latest status of these patients. We also did not access records of SREs during the period of one year before the index date, to rule out the influence of previous history of SRE on the outcomes. We used medication, diagnosis, and lab tests 1 year preceding the index date as the input (this period had no SREs). For lab tests, we only included abnormal ones in our modeling by searching those RESULT_FLAG labeled as “ABNORMAL”, “HIGH”, or “LOW”. We also excluded those lab tests with low frequency and kept the 709 most frequently tested ones. The diagnosis was coded in ICD9 before 2015 and ICD10 after 2015. As such, we used a lookup table from https://www.cms.gov/Medicare/Coding/ICD10/2018-ICD-10-CM-and-GEMs (accessed on 23 March 2021) to convert ICD9 to ICD10 codes. The first three characters of the ICD10 codes that designate the category of the diagnosis were extracted, yielding 1639 diagnosis groups. Medication names were converted to Drugbank IDs by name matching, and 1421 unique Drugbank IDs were mapped. Finally, for each encounter, the associated medications, diagnosis, and abnormal lab test results were packed into a sequence with the indices of Drugbank IDs, categories of the diagnosis, and lab test IDs, respectively.

### 2.3. Data Augmentation

Data augmentation is a technology used to increase the data size and to reduce overfitting [34,35]. It has been widely used in deep learning approaches for image classification. For example, researchers can translate, rotate, and resize the original images to generate new images for training [36]. In our study, to increase the sample size of patients with SREs (case group), we oversampled data around the SRE events. This allowed us to obtain multiple datapoints from one patient with different information collected from different encounters. At any encounter, the chance of having SREs within the next three months were much lower than having no SREs, even for PTSD patients at high risk. We included all encounters within three months prior to the SREs, which satisfied the inclusion criteria for cases. Meanwhile, to balance the amounts of data for the case and control groups, we under-sampled the encounters, which satisfied the inclusion criteria for controls, so the results of the study would not be biased by the number of subjects in the two groups. The purpose of data augmentation is to enhance the influence of factors nearby the events while reducing the effects of factors far from the events.

### 2.4. Dataset Splitting

The dataset was split with a ratio of 8:1:1, and 8 of 10 subsets were used as the training dataset, while 1 of 10 subsets was used as the validation dataset to find the optimal parameters; the remaining 1 subset was used as the test set to evaluate the generalization of our model. In the machine/deep learning field, this is a well-accepted approach to train models and avoid overfitting the data [37].

### 2.5. DeepBiomarker

We adopted the Pytorch_EHR framework established by the Zhi Group, where Deep learning models with Vanilla RNN, LSTM, Bidirectional RNN, Bidirectional LSTM, Dilated RNN [38], Dilated LSTM, QRNN [39], and T-LSTM were used to analyze and predict clinical outcomes [30]. The Zhi Group has access to multiple EHR databases with over 50 million patients, and their approaches are designed for uncovering the logic of medical practice and helping improve the efficiency of clinical care. In addition to their algorithms, we further modified the framework as highlighted in Figure 1 by (a) data augmenting to improve the model performance; (b) including individual lab tests and medications along with the diagnosis groups as the input, so that we could assess the effects of each lab tests and medications; and (c) integrating a contribution analysis [31] module for the importance estimation of key factors (see below for more details). The structure we used was the LSTM model, which stores previous illness history, infers current illness states, and predicts future medical outcomes [40].

The memory cell is gated to moderate the information flow to or from the cell. In this study, the following parameters were used: embed dimension: 128; hidden size: 128; dropout rate: 0.2; number of layers: 2; input size: 30,000; patience: 3. The calculations were repeated ten times for each deep-learning algorithm to estimate the standard deviations of the accuracy.

### 2.6. Assessment of Importance of the Clinical Factors for Predicting Suicide-Related Events

To further investigate the importance of these factors on the prediction of SRE, we calculated the relative contribution (RC) of each feature on the SRE [31]. The RC of a feature was calculated as the average contribution of the feature to events, divided by the average contributions of this feature to no-events. The contributions were estimated by a perturbation-based approach. Such an approach has been used in recent study on the important features for heart failure incidence prediction [41]. The equation is shown as follows, where *FC* represents the feature contribution:(1)$$RC=\frac{\frac{1}{m}\;\sum FC_{with\;event}}{\frac{1}{n}\;\sum FC_{without\;event}}$$

*FC* value was the total value of the feature within the same patient if the feature appeared more than once in that patient, *m* and *n* are number of patients with and without event, respectively. The natural logarithm form variance for *RC* was calculated as:(2)$$Variance(\ln(RC))=\frac{{\left(\frac{sd\;of\;FC\;of\;patients\;with\;event}{mean\;of\;FC\;of\;patients\;with\;event}\right)}^{2}}{number\;of\;patients\;with\;event}+\frac{{\left(\frac{sd\;of\;FC\;of\;patients\;without\;event}{mean\;of\;FC\;of\;patients\;without\;event}\right)}^{2}}{number\;of\;patients\;without\;event}$$
where *sd* is standard deviation.

Thus, the 95% confidence interval (*CI*) of *RC* is given by
(3)$$95\%CI=e^{(\ln\left(RC\right)\pm 1.96\sqrt{Variance\left(Ln\left(RC\right)\right)}}$$
and the *p*-value is under the assumption of z distribution [42]. Bonferroni correction [43] was used to reduce the type I error caused by multiple comparison.

### 2.7. Assessment of Model Performance

Model performance was evaluated by the area under the ROC curve (AUROC).

## 3. Results

### 3.1. The Performance of DeepBiomarker on SRE Prediction

We identified 38,807 PTSD patients from UPMC EMR data. We further identified 11,695 cases and 11,695 controls from patients with more than 1 year of EMRs before the diagnosis of PTSD. These samples were split to an 8:1:1 ratio for training, validation, and test sets. The performance of the DeepBiomarker with different deep-learning algorithms and features can be found in Table 1.

As shown in Table 1, the RNN, DRNN, QRNN, LSTM, and RETAIN algorithms implemented in DeepBiomarker all showed excellent performance on SRE prediction, i.e., all yielded equal to or more than 0.90 of AUROC, with no significant difference among them. In addition, we tested whether deducting the data types used in the model would greatly impact the model, and though the AUC reduced using diagnosis data only, the decrement was slight and the performance of using diagnosis data only was also excellent (greater than 0.90). This suggests that our model can not only accurately predict the SRE but can adapt to various data availability, so it can be applied in broader settings. On the other side, our results showed that lab tests did provide additional information for improving the performance of our models.

### 3.2. Important Indicators for SRE Prediction

As mentioned above, we used a perturbation-based estimation to calculate the relative contribution of each feature on the prediction of SRE. The following features were identified to have significant impact on SRE predictions: glucose, glucose urine, glucose (bedside), chloride, red cell distribution width (RDW), mean corpuscular hemoglobin (MCH), hemoglobin (HGB), hematocrit, mean corpuscular volume (MCV), white blood cell (WBC), anion gap, blood in urine, absolute (ABS) neutrophils, absolute (ABS) basophils, potassium, international normalized ratio (INR), ionized calcium, ionized calcium instrument description and testing technologies (ISTAT), calcium, bacteria, base excess, total protein, sodium, prothrombin time, mean platelet volume (MPV), and platelets. Table 2 lists the lab tests that were identified as important features in the DeepBiomarker model. A feature was included in the list if the *p* value (Bonferroni adjusted) < 0.05 and its confidential interval did not contain 1. These identified factors imply the possibility of regulation of the pro-inflammatory system, immune system, energy metabolism, acid-base balance, oxidative stress, respiratory system, cardiovascular system, blood circulatory system, trauma, and gut bacteria, which shape the pathophysiological pathways promoting depression and suicidal risks in PTSD patients. Table 3 and Table 4 list the medications and diagnoses that were identified as important features in the DeepBiomarker model, respectively. A feature was included in the list if the *p* value (Bonferroni adjusted) < 0.05 and its confidential interval did not contain 1. The full list of the important features can be found in Appendix A: Important features identified by perturbation-based contribution analysis for SRE prediction.

### 3.3. Effect of Medication use and Comorbidities on PTSD for SRE Prediction

Physicians generally prescribe epinastine to patients with allergic conjunctivitis (Table 4) and allergic rhinitis [44] to prevent itching. Epinastine does not penetrate the blood-brain barrier and, therefore, is not expected to induce side effects of the central nervous system. We investigated the diagnosis of patients who had ever used epinastine and found 11 of 31 these patients had the diagnosis of allergic rhinitis. However, studies have shown that antihistamines may cross the blood brain barrier due to polarity and cationic charge at the PH of the physiological state, thus leading to CNS depression and suicidal tendencies in patients with allergic rhinitis. It is possible that epinastine may promote CNS depression and SREs in these patients [45,46]. We could not rule out the possibility that these epinastine users might have an active allergy status. Hypothyroidism and osteoporosis (Table 4) are common co-morbidities among women that are linked to increased risk of PTSD [47,48,49]. A case study examined the excessive use of diltiazem (or CARTIA in Table 3), a calcium channel blocker promoting depression, suicide, and rhabdomyolysis in these patients [50,51,52]. Another study showed the possibility of Hashimoto’s encephalopathy for depressed patients being prescribed diltiazem [53]. Torsemide is a diuretic medication used to treat fluid overload due to risk of heart, kidney, and liver disease (Table 4). Our DeepBiomarker predicted that Torsemide can increase the risk of SREs. Two possible reasons can explain this. Torsemide use may be indictive of severe disease status, or it inhibits the Na+/K+/Cl- pump and thus exacerbates existing imbalances of sodium, potassium, and chloride ions. Unconventional therapies like ustekinumab (or STELARA in Table 3) might promote PTSD and SREs among severe Crohn’s disease patients (Table 4) [54,55]. Conventional medications like rosuvastatin are linked to psychiatric adverse effects due to abnormal cholesterol levels [56]. In conjunction with these medications, other comorbidities like migraine, epilepsy, and hepatitis C may interact with the inflammatory system and lead to the development psychiatric symptoms in patients due to the overlap of underlying mechanistic pathways existing between them [57,58].

Our results also show that some medications may be protective towards SREs in PTSD patients. Glucocorticoids are the most widely used anti-inflammatory therapy for a variety of immunological disorders. Some studies show higher doses of glucocorticoids can cause temporary alterations to the neuroplasticity of the hippocampus. This mechanism may contribute to hypomania, though evidence is inconsistent. Recent literature confirms that the likelihood of inducing psychiatric symptoms follows a dose response correlation [59,60]. One study noted that patients who received 40 mg/day of prednisolone had hypomania compared to patients who received lower doses [61]. Conversely, current evidence indicates that glucocorticoids may protect against the development of PTSD without risks of SREs [59]. Vitamin deficiencies also promoted psychiatric symptoms, such as altered memory, cognitive impairment, dementia, depression, and suicide (Table 4). A recent study showed a negative correlation between deficiency of Vitamin K, Vitamin B12, Vitamin D, and other multivitamins and depression in higher BMI groups [62,63,64]. Folate and Vitamin B12 deficiencies lead to macrocytic anemia and MDD [62,63,64]. Another study showed depressed colonoscopy patients using Suprep showed no microbiome dysbiosis and depression [65]. Thus, the integrative use of Suprep can be extended to PTSD patients, to maintain a healthy brain–gut connection without exacerbation of SREs. However, future studies are necessary to substantiate this claim. Apart from BMI [66,67,68], hair loss is also indicative of poor quality of life. Saw Palmetto is a treatment option for patients with androgenetic alopecia and self-perceived hair thinning. This would help in improving depression along with quality of life [69]. Apart from nutrition, immune status also influences mental health. Immunosuppressive drugs like tacrolimus, mycophenolate, and mycophenolic acid could be used for PTSD patients with end-stage renal disease, without resultant SREs [70,71]. Other drugs such as cephalexin, tobramycin-dexamethasone, and fosfomycin may be protective or risky [72]. We also found that pregnancy could be a surprising protective factor, because most women expect that a baby will bring more joy and motivation to their lives (Table 4). Our results are in line with a trajectory analysis done in 319 women, where more than half of the women had high PTSD symptoms early on, but their symptoms declined later on in their pregnancy [73]. This shows that a combination of clinical support and treatment would help alleviate PTSD.

### 3.4. Overall Lab Test-Based Indicators of Comorbidities and Disease Burdens for SRE Prediction

Through further analysis on the DeepBiomarker model, we identified the important lab tests as the biomarkers. Depending upon the type of biomarker, they are classified into blood, kidney, metabolic syndrome, cardiovascular, urine, and miscellaneous (Table 5). These biomarkers are strongly correlated to depression (depression itself has Bonferroni adjusted *p*-value 2.80 × 10^−86^) and suicide along with its implications on adjoining diagnoses and medication use. These lab tests are indicators of underlying comorbidities and thus can be considered as measurements of disease burdens. It is not a surprise that most of these lab tests increase the risk of SREs. However, a few of them did provide protective effects, which need to be interpreted with caution. For example, abnormal levels of cholesterol are predicted as an indicator of reduced risk of developing SREs. We analyzed the lab test records of cholesterol and found that most of these lab tests (13,515 of 14,167) show high cholesterol levels, while low cholesterol has been reported to increase risk of suicide [74]. The difference between ionized calcium and ionized calcium ISTAT may also be explained, because we found fewer “low” flags (87/136 = 0.64) in the ionized calcium results than in the ionized calcium ISTAT (1643/2218 = 0.74) and calcium (CA) (88844/95294 = 0.93); lower calcium level may increase the risk of suicide [75]. In addition, patients with head injury might suffer from central nervous depression due to respiratory acidosis (accumulation of carbon dioxide in the body). This base excess reactivity could be also used as a prophylactic biomarker [76]. However, research studies have shown that base excess is a superior biomarker in other co-morbidities, such as septic shock and ICU mortality [77,78].

## 4. Discussion

In this retrospective study, we built a deep learning model, DeepBiomarker, to predict the SREs based on the abnormal results of regular lab tests in the last year, together with the diagnosis and medications used in the same period. The model yielded very good performance, with an AUC score above 0.930, which is better than the traditional machine learning models such as decision tree and random forest (See Appendix A: Performance of traditional machine learning algorithms). The improvement might be from the fact that DeepBiomarker can also consider the time effects of these features. For the sake of discussion, we grouped our top biomarkers specific to PTSD based on their type.

### 4.1. Biomarkers Closely Related to PTSD and SREs

**Blood-based biomarkers**. We identified leading blood-based biomarkers that are potentially useful for assessing the risks of SREs in PTSD patients. Many studies have suggested blood-based biomarkers are associated with mental disorders. A study showed PTSD patients with antidepressant therapy had decreased platelet counts, mean platelet volume (MPV), and neutrophil-lymphocyte ratio (NLR) compared to controls [80,86]. Another study found higher NLR and MPV in violent attempters compared to non-violent attempters [83].

**Kidney diseases.** Hypochloremia potentiates metabolic acidosis in the initial stages and high anion gap acidosis in the later stages of CKD [93]. To the best of our knowledge, we are the first to propose chloride and anion gap as a possible biomarker for PTSD and SREs. Another novel biomarker that could possibly influence SREs is sodium. Abnormal sodium levels are reportedly associated with suicide attempters [96,97]. Current research has yet to elucidate a deeper relationship between these ion biomarkers and suicidal patients with renal co-morbidities, but we are optimistic that our tool correlated novel renal biomarkers such chloride, anion gap, and sodium, with unique diagnosis and medication patterns.

**Metabolic syndrome.** Metabolic syndrome refers to a cluster of metabolic conditions like dyslipidemia, insulin resistance, and hypertension leading to heart diseases. There are studies examining the role of PTSD in metabolic syndrome directly [112,113], such as Heppner et al., who found a significant association between PTSD severity and increased likelihood of metabolic syndrome [114]. Current research hypothesizes that abnormal cholesterol levels and glucose levels in depressed patients might be due to low serotonin and higher aggression and suicide levels [56,100]. However, it is important to thoroughly examine the association between PTSD severity and metabolic biomarkers in these high-risk patients.

### 4.2. New Hypothesis on SREs in PTSD

We would like to shed some light on some unique biomarkers and hypotheses that may serve as important indicators of diseases that gravely impact quality of life. SREs may emerge due to an amalgamation of biological and environmental factors. Traumatic brain injury (TBI) patients undergo a coagulation dysfunction after a fall or injury due to abnormal prothrombin time, potassium, and INR.

This is also seen in atrial fibrillation, recurrent myocardial infarction, and transient ischemic attack [105]. In addition, individuals with obstructive sleep apnea (OSA) and PTSD have high hemoglobin and hematocrit [87,115,116]. Once the coagulation dysfunction cascade is activated, a prothrombotic state emerges, due to fibrinolysis shutdown and hyperactive platelets [85,117]. While the overall role of the coagulation system is unclear, the primary regulators are platelet dysfunction, endothelial activation, disturbed fibrinolysis, and inflammation [118]. Inflammation also contributes to serotonergic, noradrenergic, and dopaminergic dysfunction, which lead to alterations in the brain [119,120]. MDD and inflammatory bowel disease (IBD) patients diagnosed with gut microbiome dysbiosis have abnormal pro-inflammatory cytokine levels [110]. These cytokines affect the neural activity of the brain via the HPA (hypothalamus pituitary adrenal) axis. Other disorders of the stomach and duodenum, esophageal reflux, and periumbilical pain are also exacerbated due to microbiome dysbiosis; it would be interesting to examine the psychiatric symptoms in these patients [111]. Another unconventional hypothesis that could be extrapolated and used for future studies is correlating allergies to PTSD and suicidal behavior. A study showed pollen-specific immunoglobulin-E and WBCs to be associated with worsening of depressive scores in bipolar patients during high pollen seasons. In addition, PTSD patients with nicotine dependence and chronic obstructive pulmonary disorder (COPD) had high RBCs and suicide attempts compared to non-smokers [79]. Based on our results, we show that activation of inflammatory mediators due to asthma and allergy rhinitis may be potential biomarkers for predicting SREs in PTSD.

### 4.3. Biomarkers for Personalized Treatment for PTSD to Reduce the Risk of SRE

Adapting the multiple biomarker diagnosis model for nervous system diseases can provide more conducive and personalized assessment of PTSD patients with SREs [65]. Using EMR data, we monitored multiple biomarkers with respect to patient medication history and diagnosis to propose newer therapies for PTSD patients with SREs. The literature also emphasizes the need for real-time evaluation of psychological variables to better predict suicide risks in PTSD patients with SREs. The existence of many variables that were tracked in real time in our study may refine evidence pertaining to the existence of a multi-casual and equifinal etiology of PTSD coupled with SREs, which proposes many interchangeable contributing factors and casual pathways. Our findings augment prior work by translating multiple arrays of information from EMR data into a versatile predictive model that can develop logical approaches that would provide easy interpretation of these multiple biomarkers to be applied to other fields of study. From a practical standpoint, such multiplicity points to the potential superiority of data-informed deep learning prediction tools for future SRE risk assessments. Our study is superior because of its multi-faceted approach, sample size, time effect consideration, convenience, low cost, and suitable choice of routine testing in a clinical setting. Unlike current biomarker studies, we considered all (both clinically and non-clinically applicable) biomarkers in our study. This emphasizes the ability of deep learning to not reject less frequently recorded or less predictive features and thereby completely extract and consider all the information necessary for providing the most clinically applicable biomarkers from our data set. One of the many future uses of DeepBiomarker is as a decision support algorithm in a clinical setting, to progressively enrich the positive impact of a data-informed algorithmic approach for personalized risk assessment.

## 5. Limitation of Our Study

Our study has a few limitations: First, there could be inconsistencies in biochemical test results between patients due to enrollment bias, and some lab tests might have low representation in our database. As such, the analysis might have limited power to detect the effects. Second, we used EMR data from January 2004 to October 2019, and in this period, there is a possibility of changes in treatment and number of lab tests among these patients. However, these limitations are caused by using EMR as a data source and can be resolved by investigations using a randomized clinical trial or prospective design. Third, we considered the effect of biomarkers along with diagnosis and medication use; however, in our results, comorbidities had a higher impact compared to biomarkers. This can be explained, since diagnosis considers the past status of the patients while biomarkers take into consideration the recent status of PTSD and SREs. In addition, our analysis did not consider the etiology of PTSD, which is very challenging to access from the EMR data. Linking our clinical findings to the current well-known biomarkers and brain structure changes warrants further investigation.

## 6. Conclusions

Our deep learning approach aimed to identify biomarkers for the assessment of the risk of SREs among PTSD patients. We found medications like prednisone, vitamin supplements (vitamin D, mephyton (vitamin K), vol-plus (multivitamin) and vitamin B-12)), tacrolimus, mycophenolate, mycophenolic acid, cephalexin, tobramycin-dexamethasone, monurol (Fosfomycin), saw palmetto, and suprep bowel to have the potential to reduce the risk of SREs among PTSD patients. We would like to emphasize the importance of using these biomarkers and medications in a context-dependent manner, coupled with the right diagnosis, to obtain superior clinical outcomes. Our findings support the existence of multiple biomarkers that equally and exhaustively predict SRE risks in PTSD patients. DeepBiomarker has great potential to receive wider clinical access and can be complementary to current established biomarkers and changes in brain structure, brain volume, and neural circuits to develop more transparent population screening and newer biological hypothesis surrounding PTSD and SRE etiology. The high accuracy and versatility of DeepBiomarker, along with its capacity to accommodate multimodal information, offers a holistic approach for personalized prediction of risk assessment.

## Figures and Tables

**Figure 1 jpm-12-00524-f001:**
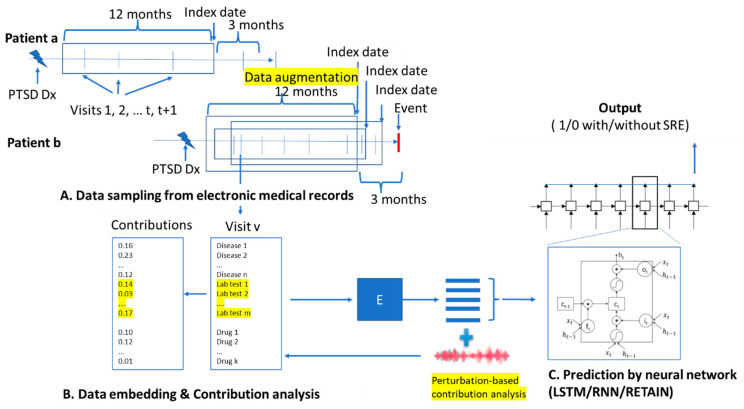
The overview of DeepBiomarker. (**A**) Data sampling from electronic medical records, (**B**) data embedding, and (**C**) prediction by neural network, with LSTM as the basic prediction units. Perturbation-based contribution analysis was used to identify important features. RNN: recurrent neural network; LSTM: long short-term memory; RETAIN: REverse Time AttentIoN model. PTSD: post-traumatic stress disorder; Dx: diagnosis; SRE: suicide-related event.

**Table 1 jpm-12-00524-t001:** The performance of DeepBiomarker with different deep-learning algorithms and features.

Whole Dataset (Diagnosis + Medication + Lab Test)				
	Valid AUC	Test AUC	Valid AUC std	Test AUC std
RNN	0.921	0.931	0.002	0.003
DRNN	0.929	0.930	0.002	0.001
QRNN	0.933	0.929	0.001	0.001
TLSTM	0.933	0.929	0.001	0.001
RETAIN	0.935	0.932	0.001	0.002
**Dataset without lab tests (diagnosis + medication)**				
	Valid AUC	Test AUC	Valid AUC std	Test AUC std
RNN	0.915	0.912	0.002	0.002
DRNN	0.917	0.913	0.001	0.002
QRNN	0.918	0.91	0.001	0.001
TLSTM	0.920	0.910	0.001	0.001
RETAIN	0.920	0.910	0.001	0.002
**Dataset without lab tests and medication (diagnosis only)**				
	Valid AUC	Test AUC	Valid AUC std	Test AUC std
RNN	0.919	0.911	0.001	0.002
DRNN	0.92	0.909	0.001	0.002
QRNN	0.919	0.908	0.002	0.002
TLSTM	0.919	0.910	0.001	0.001
RETAIN	0.919	0.910	0.001	0.002

Notes: AUC: area under curve; std: standard deviation; RNN: recurrent neural network; DRNN: dilated recurrent neural networks; QRNN: quasi-recurrent neural network; LSTM: long short-term memory; RETAIN: REverse Time AttentIoN model.

**Table 2 jpm-12-00524-t002:** Important lab test results identified by perturbation-based contribution analysis for SRE prediction.

Feature Name	Relative Contribution	CI95up	CI95down	FDR_Q	*p* Value (Bonferroni Adjusted)
Glucose	1.45	1.51	1.38	9.19 × 10^−52^	7.35 × 10^−51^
Chloride	1.41	1.49	1.35	1.58 × 10^−36^	2.21 × 10^−35^
RDW	1.52	1.63	1.42	2.38 × 10^−30^	3.81 × 10^−29^
MCH	1.50	1.61	1.40	3.55 × 10^−27^	7.45 × 10^−26^
HGB	1.39	1.47	1.32	9.40 × 10^−27^	2.16 × 10^−25^
HCT	1.38	1.46	1.31	2.54 × 10^−26^	6.36 × 10^−25^
MCV	1.50	1.61	1.39	1.52 × 10^−23^	4.56 × 10^−22^
WBC	1.37	1.45	1.29	9.61 × 10^−23^	3.08 × 10^−21^
Anion Gap	1.44	1.55	1.34	1.45 × 10^−18^	5.66 × 10^−17^
Blood−Urine	1.29	1.35	1.22	1.51 × 10^−18^	6.05 × 10^−17^
ABS Neutrophils	1.29	1.36	1.22	1.48 × 10^−17^	6.21 × 10^−16^
Glucose−Urine	1.69	1.89	1.51	6.40 × 10^−17^	2.82 × 10^−15^
Glucose (Bedside test)	1.38	1.49	1.28	6.23 × 10^−15^	3.43 × 10^−13^
ABS Basophils	1.35	1.45	1.26	2.62 × 10^−14^	1.60 × 10^−12^
RBC	1.27	1.34	1.20	3.61 × 10^−14^	2.24 × 10^−12^
Potassium	1.28	1.35	1.20	7.41 × 10^−14^	4.81 × 10^−12^
INR	1.96	2.31	1.66	4.61× 10^−13^	3.32 × 10^−11^
Ionized Calcium, ISTAT	1.66	1.88	1.46	1.65 × 10^−12^	1.24 × 10^−10^
Ionized Calcium	0.498	0.598	0.415	9.33 × 10^−12^	7.47 × 10^−10^
Bacteria	1.23	1.30	1.16	9.21 × 10^−11^	8.01 × 10^−9^
Base excess	0.460	0.573	0.370	3.80 × 10^−10^	3.57 × 10^−8^
Total protein	1.24	1.32	1.17	1.16 × 10^−9^	1.10 × 10^−7^
Calcium	1.20	1.27	1.14	1.32 × 10^−9^	1.27 × 10^−7^
Red blood cells−Urine	1.22	1.29	1.15	2.09 × 10^−9^	2.03 × 10^−7^
Sodium	1.26	1.35	1.18	2.67 × 10^−9^	2.67 × 10^−7^
Prothrombin Time	1.75	2.06	1.48	2.71 × 10^−9^	2.74 × 10^−7^
MPV	3.84	5.77	2.56	6.06 × 10^−9^	6.19 × 10^−7^
Platelets	1.29	1.39	1.19	1.03 × 10^−8^	1.10 × 10^−6^

Notes: RDW: red cell distribution width, MCH: mean corpuscular hemoglobin, HGB: hemoglobin, HCT: hematocrit, MCV: mean corpuscular volume, WBC: white blood cell, ABS neutrophils: absolute neutrophil count, ABS basophils: absolute basophil count, RBC: red blood cell, INR: international normalized ratio, ionized calcium ISTAT: ionized calcium instrument description and testing technologies, MPV: mean platelet volume. FDR_Q: false discovery rate adjusted Q value; CI: confidence interval.

**Table 3 jpm-12-00524-t003:** Important medication use results identified by perturbation-based contribution analysis for SRE prediction.

Feature Name	Relative Contribution	CI95up	CI95down	FDR_Q	*p* Value (Bonferroni Adjusted)
Epinastine HCL 0.05% Eye drops	2.76	3.02	2.532	6.09 × 10^−99^	1.83 × 10^−98^
Vitamin D2 1.25 mg (50,000 Unit)	0.673	0.740	0.611	1.33 × 10^−13^	9.19 × 10^−12^
Tacrolimus Cap 1 mg	0.415	0.520	0.332	3.63 × 10^−12^	2.76 × 10^−10^
Mephyton 5 mg tablet	0.248	0.361	0.171	3.39 × 10^−11^	2.88 × 10^−9^
Mycophenolate 250 mg capsules	0.330	0.449	0.243	1.89 × 10^−10^	1.72 × 10^−8^
Cartia XT (Diltiazem) 240 mg Capsule	1.82	2.18	1.53	2.10 × 10^−9^	2.06 × 10^−7^
Cephalexin 500 mg Capsule	0.816	0.869	0.767	8.62 × 10^−9^	8.97 × 10^−7^
Vol−plus tab (Multivitamin)	0.794	0.852	0.739	9.33 × 10^−9^	9.80 × 10^−7^
Stelara (Ustekinumab) 90 mg/mL syringe	3.05	4.30	2.17	1.02 × 10^−8^	1.08 × 10^−6^
Monurol (Fosfomycin) 3 gm Sachet	0.344	0.479	0.247	1.78 × 10^−8^	1.98 × 10^−6^
Trospium Chloride ER 60 mg Capsule	2.25	2.91	1.74	2.70 × 10^−8^	3.13 × 10^−6^
Vitamin B12 500 mcg Tablet	0.755	0.826	0.690	4.03 × 10^−8^	4.71 × 10^−6^
Saw Palmetto 160 mg Capsule	0.403	0.540	0.300	6.10 × 10^−8^	7.32 × 10^−6^
Mycopgenolic Acid 180 mg Tb	0.295	0.443	0.197	1.46 × 10^−7^	1.93 × 10^−5^
Prednisone tablet 10 mg	0.835	0.886	0.786	1.60 × 10^−7^	2.12 × 10^−5^
Suprep Bowel Prep Kit	0.848	0.897	0.802	2.61 × 10^−7^	3.63 × 10^−5^
Zenatane (Isotretinoin) 20 mg Capsule	3.77	6.06	2.35	1.24 × 10^−6^	0.000185
Tobramycin-dexamethasone Suspension	0.665	0.771	0.573	2.06 × 10^−6^	0.000314
Torsemide 10 mg tablets	3.71	5.98	2.30	2.34 × 10^−6^	0.000360
Rosuvastatin 10 Mg Tablet	1.71	2.10	1.40	4.42 × 10^−6^	0.000730

Notes: Relative contribution value > 1: risk; relative contribution value < 1: protective; FDR_Q: false discovery rate adjusted Q value; CI: confidence interval.

**Table 4 jpm-12-00524-t004:** Important diagnosis results identified by perturbation-based contribution analysis for SRE prediction.

Feature Name	Relative Contribution	CI95up	CI95down	FDR_Q	*p* Value (Bonferroni Adjusted)
Esophageal reflux	1.72	1.79	1.65	1.10 × 10^−128^	2.20 × 10^−128^
Major depressive disorder, single episode, unspecified	1.58	1.65	1.52	6.99 × 10^−87^	2.80 × 10^−86^
Nicotine dependence, cigarettes, uncomplicated	1.52	1.58	1.46	1.58 × 10^−80^	7.90 × 10^−80^
Personal history of transient ischemic attack (TIA), and cerebral infarction without residual deficits	1.82	1.93	1.72	2.21 × 10^−77^	1.33 × 10^−76^
Anxiety state, unspecified	1.50	1.56	1.44	7.55 × 10^−74^	5.29 × 10^−73^
Periumbilical pain	1.88	2.04	1.74	5.25 × 10^−49^	4.72 × 10^−48^
Bipolar disorder, unspecified	1.74	1.86	1.62	6.88 × 10^−46^	6.88 × 10^−45^
Unspecified asthma with (acute) exacerbation	1.49	1.58	1.42	2.51 × 10^−43^	2.76 × 10^−42^
Epilepsy, unspecified, without mention of intractable epilepsy	1.86	2.03	1.71	2.08 × 10^−41^	2.50 × 10^−40^
Migraine, unspecified, without mention of intractable migraine without mention of status migrainosus	1.53	1.63	1.45	5.85 × 10^−41^	7.61 × 10^−40^
Irritable bowel syndrome	1.89	2.09	1.72	5.50 × 10^−32^	8.25 × 10^−31^
Mononeuritis of unspecified site	1.45	1.54	1.37	5.31 × 10^−30^	9.03 × 10^−29^
Unspecified essential hypertension	1.73	1.89	1.58	2.36 × 10^−29^	4.25 × 10^−28^
Personal history of tobacco use	1.33	1.40	1.27	4.27 × 10^−28^	8.11 × 10^−27^
Anemia, unspecified	1.44	1.53	1.36	1.15 × 10^−27^	2.29 × 10^−26^
Unspecified viral hepatitis C without hepatic coma	2.06	2.32	1.82	5.20 × 10^−27^	1.14 × 10^−25^
Supervision of normal first pregnancy	0.459	0.523	0.402	9.56 × 10^−27^	2.29 × 10^−25^
Other specified disorders of stomach and duodenum	2.45	2.86	2.10	1.86 × 10^−25^	4.83 × 10^−24^
Osteoporosis, unspecified	2.45	2.88	2.09	5.31 × 10^−24^	1.43 × 10^−22^

Notes: Relative contribution value > 1: risk; relative contribution value < 1: protective; FDR_Q: false discovery rate adjusted Q value; CI: confidence interval.

**Table 5 jpm-12-00524-t005:** Top biomarkers for prediction of suicide-related events (SREs) in PTSD patients along with the effect on depression, suicide, related diagnosis, and medication use.

Lab Test	Effect on Depression	Effect on Suicide	Diagnosis	Medication Use
**Blood:**				
Auto Absolute Basophils	High EBR (Eosinophil to Basophil ratio) [79]	Poor quality of life, and suicide attempts [79]	Chronic Obstructive Pulmonary Disease (COPD) and asthma [79]	
Mean corpuscular hemoglobin concentration (MCHC)	Macrocytic anemia associated with folate and B12 deficiencies [62,63,64]		Anemia [62,63,64]	
Red blood cell (RBC)	Increased circulating blood cells [80]		Chronic Obstructive Pulmonary Disease (COPD) [81] and nicotine use and dependence [82]	
Neutrophils	High Neutrophil Lymphocyte ratio [79]	Higher NLR in violent suicide attempters [83]	Migraine, Epilepsy [57]	
Mean Platelet Volume (MPV)	Increased mean platelet volume (MPV) [79]	Higher MPV in violent suicide attempters [83]		
Platelets	High platelet count [80]	Higher MPV in violent suicide attempters [83]	Hepatitis C [58] and iron deficiency [84]	
Prothrombin time (PT)	Abnormal prothrombin time (aPTT) in traumatic brain injury (TBI) patients [85]			
Auto Absolute Lymphocytes	High NLR [86]	PTSD and COPD: poor quality of life, and suicide attempts [79]	COPD and asthma [79]	
Mean corpuscular hemoglobin (MCH)	Macrocytic anemia and major depressive disorder (MDD) associated with folate and B12 deficiencies [62,63,64]		Anemia [62,63,64]	Folate and B12 deficiencies [62,63,64]
Hematocrit	Obstructive sleep apnea (OSA) and PTSD: High hemoglobin and hematocrit [87]	Higher suicidal rates among patients in higher altitudes [87]		
Red cell distribution width (RDW)	High RDW [88]	RDW is a predictor of mortality in suicidal patients with organophosphate insecticide (OPI) poisoning [89]		
Hemoglobin (HGB)	Higher hemoglobin [90]			
Mean corpuscular volume (MCV)	High MCV [90]			
White blood cell (WBC)	High WBC count in depressed patients [91]	High WBC count amongst suicide attempters [91]	Mononeuritis and Bipolar disorder [91,92]	
**Kidney:**				
Chloride	Hyperchloremia [93,94]		Chronic kidney disease (CKD) [93,94]	
Sodium	Hypernatremia in traumatic brain injury [95]	High vasopressin and sodium concentrations in depressed suicide attempters [96,97]		
Anion Gap	High anion gap acidosis [93,98]	Poor quality of life [94]	Hypertension [99]	
**Metabolic syndrome:**				
Glucose, POC/Glucose (Beside Test)	High blood glucose peaks and insulin levels in PTSD patients [100]	High blood glucose in suicide [101]		
Cholesterol	Low serum HDL-cholesterol levels are associated with long symptom duration [102]	Abnormal cholesterol levels may cause higher suicide risk [74]		
**Cardiovascular:**				
INR	Elevated INRs in depressed patients [103,104]		Atrial fibrillation, recurrent myocardial infarction, transient ischemic attack [105]	
Potassium	Hyperkalemia in depressed patients [103,104]		Atrial fibrillation, recurrent myocardial infarction, transient ischemic attack [105]	
**Urine:**				
Blood urine	Bacterial vaginosis increases blood urine content (BV) [106]	Poor quality of life of UTI patients increases risk of SREs [107]		Trospium chloride [108]
**Miscellaneous:**				
Total Protein	High total protein content in trauma patients [109]			
Gut microbiome (Bacteria)	High gut microbiome dysbiosis and inflammation in MDD [110]	Gut microbiome dysbiosis causes increased SREs [110]	Irritable bowel syndrome, Disorders of the stomach and duodenum, esophageal reflux, MDD, periumbilical pain [110,111]	Ustekinumab (Stelara) [54]
Calcium	Abnormal calcium levels, common among women, have been linked to increased risk of PTSD [47]	Abnormal calcium levels trigger the immune mediators to promote depression, anxiety, and suicidal behavior [49]	Osteoporosis and hypothyroidism, anxiety [48,49]	Diltiazem (CARTIA) [53]

## Data Availability

The data used in this study were from UPMC under a data use agreement. The authors are not permitted to distribute the data to any third party, but researchers may contact UPMC for data access.

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
