# Peer review of "DeepBiomarker: Identifying Important Lab Tests from Electronic Medical Records for the Prediction of Suicide-Related Events among PTSD Patients"

_jpm, 2022, doi:10.3390/jpm12040524_

Round 1
Reviewer 1 Report
Miranda and colleagues in the present study entitled ‘DeepBiomarker: Identifying Important Lab Tests from Electronic Medical Records for the Prediction of Suicide-related Events among PTSD Patients’, investigated the current status of knowledge of using biomarkers for assessing the risk of suicide-related events (SREs) among post-traumatic stress disorder (PTSD) patients. For this purpose, the authors decided to identify discriminable abnormal lab test results as potential biomarkers for the assessment of suicide risk in PTSD patients and developed a deep learning model (DeepBiomarker) to analyze electronic medical records (EMR) data of PTSD patients from University of Pittsburgh Medical Center. The results of this study helped identifying some important lab tests for suicide prediction, which were also involved in shaping the pathophysiological pathways promoting depression and suicidal risks in PTSD patients. The authors concluded by stating that these potential biomarkers combined with medication use and diagnosis could facilitate in predicting PTSD patients with SREs risk.
The main strength of this manuscript is that it addresses an interesting and timely question, providing a captivating interpretation and describing how potential biomarkers combined with medication use and diagnosis could facilitate in predicting PTSD patients with SREs risk. In general, I think the idea of this perspective article is really interesting and the authors’ fascinating observations on this timely topic may be of interest to the readers of the Journal of Personalized Medicine. However, some comments, as well as some crucial evidence that should be included to support the author’s argumentation, needed to be addressed to improve the quality of the manuscript, its adequacy, and its readability prior to the publication in the present form, in particular reshaping parts of the Introduction and Discussion sections by adding more evidence and theoretical constructs.
Please consider carefully the following comments:
- Abstract: According to the Journal’s guidelines, the abstract should be a total of about 200 words maximum, but the current one includes 250 words. Please, correct it.
- Introduction: The ‘Introduction’ section is well-written and nicely presented. The Authors took a rather narrow view on the current state of knowledge surrounding pathophysiology as well as the biological vulnerability of PTSD. Nevertheless, I believe that more information about dysfunctional brain circuits underlying PTSD would provide a better background here. Thus, I suggest the authors to make an effort to provide a brief overview of the pertinent published literature that offer a perspective on brain regions with altered activity and connectivity in PTSD, because as it stands, this information is not highlighted in the text. In this regard, I believe that the statement ‘PTSD…has been associated with alterations in the brain structure, brain volume, neural circuits, genetic components, inflammation, endocrinology (including neurotransmitters, neurotrophic factors, hormones, oxidative stress), and energy metabolism’ needs some necessary references. In particular, according to this sentence, I would recommend adding evidence on the implication of frontal lobe circuitry in altered fear memory and extinction features in PTSD, with a focus on abnormalities in the ventromedial region of the prefrontal cortex (vmPFC), whose smaller volume and altered activity patterns have been observed in PTSD patients: evidence from a recent study conducted on patients with a lesion in ventromedial portion of the prefrontal cortex (https://doi.org/10.1523/JNEUROSCI.0304-20.2020) revealed that the ventromedial prefrontal cortex (vmPFC) is involved in the acquisition of emotional conditioning (i.e., learning), assessing how naturally occurring bilateral lesion centered on the vmPFC compromises the generation of a conditioned psychophysiological response during the acquisition of threat conditioning (i.e., emotional learning). I would also recommend a recent theoretical review that focused on the neurobiology of fear conditioning, and discussed the distinct contributions of anterior/posterior subregions of the vmPFC in the processing of safety-threat information: here authors provided evidence for the fundamental role of this how the region in the evaluation and representation of stimulus-outcome’s value needed to produce sustained physiological responses (https://doi.org/10.1038/s41380-021-01326-4). Moreover, if they deem it appropriate, authors can also consider additional studies that have focused on this topic (https://doi.org/10.3390/ijms21072431; https://doi.org/10.1016/j.regpep.2012.08.017).
- Introduction: In according with the previously suggested literature, to support the evidence of abnormal activity patterns in the prefrontal cortex in patients with PTSD, I believe that a recent yet relevant perspective manuscript (https://doi.org/10.17219/acem/146756) might be of interest: here the focus was on providing a deeper understanding of human learning neural networks, particularly on human PFC crucial role, that might also contribute to the advancement of alternative, more precise and individualized treatments for psychiatric disorders. This will help provide valuable insight into PTSD-related alterations in frontal lobe structural integrity and functional activity.
- Data augmentation and Data splitting: In these paragraphs, the authors described how they used a machine learning model to increase the amount of relevant data in their dataset. Since it may be challenging for readers to be able to clearly comprehend how these analyses were conducted, I was wondering whether the author could include any reference to support their statements and be more specific, thus framing the high-level context of Data Augmentation and Deep Learning.
- Results: In my opinion, this section is well organized, but it is quite concise thin and states the statistical significance of findings in an excessively broad way. Thus, I believe that this section would benefit from a more detailed and precise rewriting, in order to ensure in-depth understanding and replicability of the findings.
- Discussion: In my opinion, this article would be more compelling and useful to a broad readership if the authors moved beyond and discussed theoretical and methodological avenues in need of refinement, using this evidence to suggest a path forward. In this regard, I believe that it would have been essential to explore alternative treatments for posttraumatic core symptoms and examine the implementation of new therapeutic techniques, such as Non-invasive brain stimulation, that operate to ameliorate the symptoms of mental and neurological disorders. In this regard, I would suggest evidence from recent studies that have examined NIBS efficacy in PTSD patients (see https://doi.org/10.1038/s41398-020-0851-5): in this regard, I would suggest citing a recent review (https://doi.org/10.1016/j.neubiorev.2021.04.036) that described the potential and effectiveness of non-invasive brain stimulation (NIBS) to interfere and modulate the abnormal activity of neural circuits (i.e., amygdala-mPFC-hippocampus) involved in the acquisition and consolidation of fear memories, which are altered in many mood psychiatric disorders (i.e., anxiety disorder, specific phobias, post-traumatic stress disorder or depression). Similarly, another recent manuscript illustrated the therapeutic potential of NIBS as a valid alternative in the treatment of abnormally persistent fear memories that characterized those patients with anxiety disorders that do not respond to psychotherapy and/or drug treatments (https://doi.org/10.1016/j.jad.2021.02.076). In addition to the previously mentioned literature, authors can also see these additional studies that have focused on this topic (https://doi.org/10.2174/1871527313666141130224431; https://doi.org/10.1016/j.neubiorev.2018.05.015). These findings highlight how NIBS and are a valuable tool in research and has potential diagnostic and therapeutic applications for many mood psychiatry disorders, including PTSD, depression, or anxiety.
- I think the ‘Conclusions’ paragraph would benefit from some thoughts as well as in-depth considerations by the authors, because as it stands, it lists down all the main findings of the research, without really stressing the theoretical significance of the study. Authors should make an effort, to try to explain the theoretical implication as well as the translational application of their research.
- Tables and Figures: please provide an explanatory caption for each table within the text.
Overall, the manuscript contains 1 figure, 5 tables, 2 supplementary tables, and 110 references. The manuscript might carry important value presenting how potential biomarkers combined with medication use and diagnosis could facilitate in predicting PTSD patients with SREs risk.
I hope that, after these careful revisions, the manuscript can meet the Journal’s high standards for publication.
I am available for a new round of revision of this article.
I declare no conflict of interest regarding this manuscript.
Best regards,
Reviewer
Reviewer 2 Report
This is a paper showing a learning model combining clinical and biological features capable to predict suicidal-related events in patients diagnosed with PTSD. The project is interesting, the paper is well-written and it is of interest of the clinicians. However, prior to its publication, I recommend several minor changes.
In the abstract section, the authors reported that this study was designed to identify abnormal lab test results as potential biomarkers of suicide risk in PTSD. I would rephrase it and focus on the Main Aim of the study or the project.
In the first paragraph of the introduction section, the authors are introducing the suicide phenomenon and suicidality. I would expand this section and its clinical relevance. I consider also important to introduce the topic of biomarkers in general; in affective disorders as well as in psychiatry.
After an introduction based on suicide and biomarkers, I would introduce the rates of suicide or suicidal behavior in pacients with PTSD. Afterwards, again, I would introduce which biomarkers are available in this topic.
The main aim of the study should be clarified at the end of the introduction.
All the sections and subsections of the manuscript should be numbered.
The discussion section is brief. I would expand blood-based biomarkers, kidney diseases and metabolic syndrome and comparing them witth other biomarkers to be applied to other mental health disorders.
Round 2
Reviewer 1 Report
The authors did an excellent work clarifying the questions I have raised in my previous round of review. Currently, this paper is a well-written, timely piece of research and provides a useful summary of the existing status of how potential biomarkers combined with medication use and diagnosis could facilitate in predicting PTSD patients with suicide-related events (SREs) risk. It is well researched and nicely written, with a good balance between descriptive and narrative text.
I believe that this paper does not need further revision, therefore I accept it for publication.
Thank you for your work.